# Association of Leptin in Sarcopenia and Bone Density in Elderly Women: An Observational Analysis

**DOI:** 10.3390/diagnostics15131620

**Published:** 2025-06-26

**Authors:** Dong Gyu Lee, Jong Ho Lee

**Affiliations:** 1Department of Physical Medicine and Rehabilitation, College of Medicine, Yeungnam University, Daegu 42415, Republic of Korea; 2Department of Laboratory Medicine, College of Medicine, Yeungnam University, Daegu 42415, Republic of Korea; ae4207@naver.com

**Keywords:** leptin, sarcopenia, osteoporosis, obesity, bone mineral density, aging, fracture risk

## Abstract

**Background**: Sarcopenia and osteoporosis are common age-related conditions that markedly increase fracture risk and morbidity in the elderly. Leptin, an adipokine secreted by adipose tissue, has been implicated in musculoskeletal health, but its clinical relevance in aging populations remains uncertain. This study aimed to evaluate the associations between serum leptin levels, skeletal muscle mass, muscle strength, bone mineral density (BMD), and fracture risk in elderly women. **Methods**: This observational analysis included 79 community-dwelling women aged 65 years and older. Participants underwent assessments of body composition, serum leptin concentration, grip strength, and femoral neck BMD. Sarcopenia and obesity were classified based on established criteria. Correlation analyses and binomial logistic regression were performed to examine the relationships among leptin levels, musculoskeletal parameters, and fracture occurrence. **Results**: Leptin concentrations were significantly associated with fat-related parameters, including BMI, fat index, and total body fat percentage, but showed no significant correlation with skeletal muscle mass (ASM), grip strength, or BMD. Obese participants demonstrated higher leptin levels and fat parameters compared with non-obese participants, but no significant differences were observed in grip strength or BMD. Binomial logistic regression analysis identified femoral neck BMD and grip strength as significant independent predictors of fracture risk, whereas leptin and ASM were not identified as such. **Conclusions**: In elderly women, serum leptin levels primarily reflect adiposity rather than musculoskeletal health. Leptin is not an independent predictor of spinal fracture risk. These findings highlight the critical importance of maintaining bone density and muscle strength for fracture prevention in aging populations.

## 1. Introduction

The global trend of population aging has led to a rise in age-related conditions such as sarcopenia and osteoporosis [1]. Sarcopenia, characterized by the progressive loss of skeletal muscle mass, strength, and function with age, increases the risk of falls, fractures, disability, and reduced quality of life in older adults [2]. Osteoporosis, defined by diminished bone mineral density (BMD) and heightened fracture risk, similarly contributes to greater morbidity and mortality in this population [3]. Increasing evidence underscores the close relationship between these two conditions, highlighting shared underlying mechanisms. Older women experience distinct hormonal changes, particularly the rapid decline in estrogen levels after menopause, which accelerates bone loss and alters fat distribution [4]. These gender-specific differences in hormonal milieu and metabolic responses necessitate separate investigations to better understand disease processes in and therapeutic opportunities for women. Therefore, we conducted this study with a focus on older women to investigate the relationships between leptin and changes in bone and muscle health.

Leptin, a hormone predominantly produced by adipose tissue, is well known for its role in regulating energy balance, appetite, and metabolism [5]. Beyond these traditional functions, recent studies suggest that leptin is also involved in musculoskeletal health [6]. It appears to influence skeletal muscle mass, strength, and bone density through both direct and indirect pathways, although the full mechanisms remain to be elucidated. Leptin may therefore act as a critical link between adipose tissue changes and musculoskeletal degeneration in aging individuals [7].

Sarcopenic obesity describes the coexistence of reduced muscle mass and increased fat mass [8]. As muscle mass declines with age, basal metabolic rate decreases, often resulting in positive energy balance and fat accumulation. This added adiposity promotes chronic low-grade inflammation and metabolic disturbances, which further hasten muscle breakdown and worsen sarcopenia [9]. Leptin dysregulation, common in obesity, may drive this vicious cycle by impairing muscle anabolic pathways, promoting systemic inflammation, and disrupting bone metabolism [10]. Because leptin is central to adipose tissue signaling, abnormalities in leptin activity could play a pivotal role in the development of sarcopenic obesity, contributing not only to muscle deterioration but also to bone fragility. A deeper understanding of the interactions among muscle loss, fat accumulation, and leptin signaling is essential for addressing the multifactorial nature of sarcopenic obesity and its impact on osteoporosis.

Given leptin’s complex roles in metabolism, inflammation, and musculoskeletal remodeling, further research is necessary. Pharmacological agents targeting leptin receptors are currently under development, suggesting that clarifying leptin’s role in sarcopenia and osteoporosis could offer new therapeutic opportunities. Understanding these mechanisms could enhance strategies aimed at preserving musculoskeletal health in the aging population.

This study aimed to explore the relationships among leptin, sarcopenia, and osteoporosis in older adults, contributing to a more comprehensive understanding of their interconnected pathophysiology and informing future clinical interventions.

## 2. Materials and Methods

### 2.1. Study Population

Patients who visited the spine center of our university hospital between February 2023 and November 2024 were prospectively recruited. This cross-sectional study included 79 community-dwelling older women aged 65 years and above (mean age, 78.6 ± 5.7 years).

The inclusion criteria were as follows:Women aged 65 years or older.Agreement to undergo dual-energy X-Ray absorptiometry (DEXA) for bone and body composition assessment.

The exclusion criteria included the following:
History of femoral surgery.Use of osteoporosis medications within the past year.Current dialysis for chronic kidney disease.Ongoing chemotherapy for malignancy.Severe limitation in physical activity due to encephalopathy or major musculoskeletal disorders.

This study was reviewed and approved by the Institutional Review Board of our hospital (Approval No. YUMC 2022-10-050). Written informed consent was obtained from all participants prior to enrollment.

### 2.2. Data Collection

Anthropometric measurements included body weight and height, from which body mass index (BMI) was calculated (kg/m^2^). Femur neck BMD and skeletal mass measurement were assessed using dual-energy X-Ray absorptiometry (Discovery Wi; Hologic Inc., Marlborough, MA, USA). Total body fat percentage and fat index (fat mass/height^2^) were obtained from DEXA scans. Appendicular skeletal muscle mass (ASM) was assessed and normalized for height squared to determine muscle quantity (kg/m^2^). Grip strength was measured twice using a handheld dynamometer, and the average value from the dominant hand was used in the analysis. Bone mineral density (BMD) at the hip was evaluated using DEXA. In the data collection process, participants were also assessed for vertebral fractures. Spine compression fractures were identified based on spinal X-Ray or spine MRI imaging studies, and individuals with confirmed vertebral fractures were categorized into the fracture group for subsequent analyses. Serum leptin levels were measured using a Human Leptin ELISA Kit (Millipore, Catalog No. EZHL-80SK, Merck Millipore, Billerica, MA, USA). According to the manufacturer’s documentation, the intra-assay coefficient of variation ranged from 1.4% to 4.9%, and the inter-assay coefficient of variation ranged from 1.3% to 8.6% based on the leptin concentration of the control serum samples (0.86–28.9 ng/mL). All assays were performed in duplicate according to the manufacturer’s protocol.

### 2.3. Group Classification

Participants were categorized based on BMI into two groups: “obese” and “non-obese” (normal and underweight). The definition for obesity in adults was based on a BMI ≥ 25 kg/m^2^, following established clinical guidelines [11]. Sarcopenia was defined according to the diagnostic criteria established by the Korean Working Group on Sarcopenia (KWGS) guidelines, which emphasize low muscle mass combined with reduced muscle strength or physical performance [12]. Sarcopenia was diagnosed if grip strength was less than 18 kg or if appendicular skeletal muscle mass (ASM) was less than 5.4 kg/m^2^.

### 2.4. Statistical Analysis

Descriptive statistics were calculated for all variables. Independent sample *t*-tests were performed to compare age, BMI, ASM, grip strength, and femur neck BMD between groups. Correlation analyses were conducted to examine relationships among leptin levels, BMI, fat-related parameters, muscle mass, and bone density. A *p*-value of <0.05 was considered statistically significant. All statistical analyses were performed using IBM SPSS Statistics for Windows, Version 25.0 (IBM Corp., Armonk, NY, USA). Logistic regression analyses were conducted to assess the association between fracture occurrence and serum leptin levels, ASM, grip strength, and right hip BMD. These variables were selected based on their established relevance in the literature to musculoskeletal health and fracture risk in elderly populations. The sample size was determined based on feasibility within the study period and the need to provide sufficient statistical power. Specifically, a sample size of 79 participants was estimated to achieve 80% power to detect moderate effect sizes (Cohen’s d ≈ 0.5 or Pearson’s r ≈ 0.3) at a significance level of 0.05.

## 3. Results

A total of 79 women aged 65 years and older were included in the analysis (Table 1). The mean age was 78.6 ± 5.7 years. The mean BMI was 24.2 ± 3.6 kg/m^2^, and the mean appendicular skeletal muscle mass (ASM) was 5.12 ± 0.84 kg/m^2^. The mean grip strength was 17.3 ± 5.1 kg, and the mean hip bone mineral density (BMD) was 0.69 ± 0.11 g/cm^2^. The mean serum leptin concentration was 23.1 ± 17.2 ng/mL.

Participants were categorized into sarcopenia (n = 20) and non-sarcopenia (n = 59) groups based on KWGS criteria. The sarcopenia group showed significantly lower BMI (22.8 ± 2.7 kg/m^2^ vs. 25.1 ± 3.6 kg/m^2^, *p* = 0.003), ASM (4.73 ± 0.62 kg/m^2^ vs. 5.24 ± 0.86 kg/m^2^, *p* = 0.005), and grip strength (14.7 ± 4.2 kg vs. 18.2 ± 5.2 kg, *p* = 0.003) compared with the non-sarcopenia group. Leptin concentrations tended to be lower in the sarcopenia group, although the difference did not reach statistical significance (18.1 ± 14.5 ng/mL vs. 25.2 ± 17.8 ng/mL, *p* = 0.120).

Participants were further categorized into obese and non-obese groups based on BMI criteria (Table 2). Independent samples *t*-tests revealed that fat index (*p* < 0.001), leptin concentration (*p* < 0.001), and ASM (*p* < 0.001) were significantly higher in the obese group compared to the non-obese group. However, no significant differences were observed in grip strength (*p* = 0.383) or femur neck BMD (*p* = 0.259) between the two groups. Independent sample *t*-tests were conducted to compare fat indexes, leptin levels, ASM, grip strength, and femur neck BMD between these two groups. Independent samples *t*-tests revealed that fat index (*p* < 0.001), leptin concentration (*p* < 0.001), and ASM (*p* < 0.001) were significantly higher in the obese group compared with the non-obese group. However, no significant differences were observed in grip strength (*p* = 0.383) or femur neck BMD (*p* = 0.259) between the two groups.

In the correlation analysis, leptin levels were significantly correlated with fat-related parameters such as BMI (r = 0.360, *p* = 0.011) and total fat percentage (r = 0.413, *p* = 0.003) (Table 3, Figure 1). However, leptin was not significantly correlated with skeletal muscle mass (ASM) (r = 0.135, *p* = 0.354) or grip strength (r = −0.059, *p* = 0.686). A positive but non-significant correlation was observed between leptin levels and femur neck BMD (r = 0.195, *p* = 0.084).

To further explore factors associated with fracture risk, binomial logistic regression analyses were conducted (Table 4). Among the predictors assessed, femur neck BMD demonstrated a significant association with fracture occurrence (Estimate = −10.40, *p* = 0.004). Additionally, grip strength was identified as a significant predictor (Estimate = −0.101, *p* = 0.046), suggesting that reduced muscle strength independently contributes to fracture risk. Although ASM exhibited a borderline association with fracture (Estimate = −0.834, *p* = 0.055), it did not reach conventional statistical significance. However, serum leptin concentrations, despite their biological relevance to adiposity and muscle mass, were not significantly associated with fracture risk in this cohort (Estimate = −0.0203, *p* = 0.124).

## 4. Discussion

This study aimed to investigate the relationships between leptin levels, skeletal muscle health, and osteoporosis in older women. Our findings demonstrated that leptin concentrations were significantly associated with fat-related parameters, including BMI, fat index, and total fat percentage. However, leptin levels were not significantly correlated with skeletal muscle mass (ASM), grip strength, or femur neck BMD. These results suggest that leptin primarily reflects adiposity rather than musculoskeletal health in this population.

Previous studies have indicated that higher leptin levels are associated with preserved muscle mass and bone density, particularly in middle-aged populations [13,14]. Mechanistically, leptin is known to influence bone and muscle metabolism through multiple pathways. Leptin can promote osteoblast differentiation and inhibit osteoclast activity via central and peripheral mechanisms [15,16]. It can also stimulate muscle protein synthesis and suppress proteolysis through pathways involving the JAK/STAT and PI3K/Akt signaling cascades [17]. However, in aging populations, leptin signaling becomes impaired due to receptor-level resistance, thereby diminishing its anabolic effects on musculoskeletal tissues [18]. This resistance is characterized by impaired leptin receptor signaling and increased expression of negative regulators such as SOCS-3, which disrupt downstream signaling cascades despite elevated circulating leptin levels [19]. Although leptin plays a regulatory role in muscle metabolism, recent studies have highlighted divergent outcomes in older adults. One study reported that higher leptin levels were associated with increased muscle mass but reduced muscle strength—a phenomenon possibly linked to selective leptin resistance, through which leptin’s anabolic effects on muscle function are blunted despite elevated concentrations [20]. In contrast, another cohort study demonstrated that high serum leptin was associated with an increased risk of frailty—including weakness and exhaustion—even after adjusting for fat mass, insulin resistance, and inflammation, suggesting broader systemic effects beyond adiposity [21]. These conflicting findings suggest that both impaired leptin signaling and reduced leptin availability may variably contribute to musculoskeletal decline in older adults, depending on physiological context and study population.

In our study, no significant associations were observed between leptin levels and ASM, grip strength, or BMD. Furthermore, serum leptin concentrations in the sarcopenia group were not elevated as compared with the non-sarcopenic group. This suggests that leptin resistance may not be the predominant mechanism in this cohort; rather, lower leptin availability or general frailty may underlie the observed musculoskeletal impairments. Furthermore, discrepancies among previous studies may be attributed to methodological differences—such as the use of bone mineral content (BMC) versus bone mineral density (BMD) measurements—as well as variations in participant age and ethnicity.

Participants were also categorized into obese and non-obese groups based on BMI criteria. In this analysis, the obese group exhibited significantly higher leptin concentrations, fat index, and ASM compared with the non-obese group. The elevated leptin levels observed in obese participants align with the established understanding that leptin concentrations increase proportionally with fat mass [22]. However, no significant differences were observed in grip strength or BMD between the obese and non-obese groups. This result contrasts with previous findings from Western populations, where obese individuals often demonstrate higher BMD [23]. One possible explanation is the markedly lower prevalence of severe obesity in East Asian populations compared with Western populations, resulting in insufficient mechanical loading to influence BMD [24]. These ethnic and population-specific differences may partly account for the divergent findings regarding the musculoskeletal effects of obesity.

Binomial logistic regression analyses further revealed that femur neck BMD and grip strength were significant independent predictors of fracture risk, while leptin and ASM were not. Although ASM showed a borderline association with fracture occurrence, its effect did not reach statistical significance. These findings are consistent with previous studies that have demonstrated strong associations between bone density, muscle strength, and fracture risk in older adults. Thus, our results reinforce the fact that skeletal integrity and muscle function are critical factors for fracture prevention and align with the well-established literature in this field [25,26]. Notably, leptin did not significantly influence spinal fracture risk. Although leptin may exert beneficial effects on muscle mass and bone density, its clinical contribution appears modest compared with the direct and substantial impacts of muscle strength and bone density on fracture prevention. Therefore, while leptin may play a supportive role in musculoskeletal health, it is unlikely to serve as a principal determinant of fracture risk in elderly populations.

This study has several limitations. The sample size was relatively small, and the cross-sectional design limits causal inferences. The cross-sectional design limits the ability to infer causal relationships between leptin and musculoskeletal parameters. Further longitudinal studies are warranted to clarify the temporal and mechanistic relationships among leptin, muscle function, and bone health in aging populations. Serum leptin levels were measured at a single time point, and other adipokines and inflammatory markers were not evaluated. Additionally, the study population consisted exclusively of older women from a single geographic region, which may limit the generalizability of the findings. Despite these limitations, our results provide valuable insights into the limited role of leptin as a biomarker for musculoskeletal health in older women. Future research should incorporate comprehensive inflammatory profiling and diverse, longitudinal cohorts to better elucidate the complex interplay among leptin, inflammation, muscle, and bone in aging.

## 5. Conclusions

In this cohort of older women, serum leptin levels were significantly associated with adiposity-related measures but not with skeletal muscle mass, muscle strength, or bone density. Leptin also did not predict fracture risk independently. These findings suggest that leptin primarily serves as a marker of fat mass rather than musculoskeletal health in aging populations. Muscle strength and bone density remain the most critical determinants of fracture risk, highlighting their importance in geriatric musculoskeletal assessments.

## Figures and Tables

**Figure 1 diagnostics-15-01620-f001:**
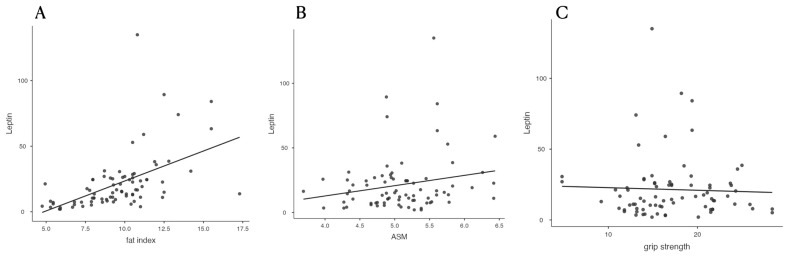
Correlation scatterplots. Scatterplots illustrating the correlations between serum leptin levels and (**A**) fat index, (**B**) appendicular skeletal muscle mass (ASM), and (**C**) grip strength among older women. Leptin levels showed a significant positive correlation with fat index but no significant correlations with ASM or grip strength. ASM, appendicular skeletal muscle mass; Fat Index, fat mass/height^2^; Leptin, serum leptin concentration; Grip strength, handgrip strength.

**Table 1 diagnostics-15-01620-t001:** Baseline characteristics of study participants.

Variable	Total (n = 79)	Sarcopenia (n = 20)	Non-Sarcopenia (n = 59)	*p*-Value
Age(year)	78.6 ± 5.7	79.7 ± 5.4	78.2 ± 5.8	0.208
BMI (kg/m^2^)	24.2 ± 3.6	22.8 ± 2.7	25.1 ± 3.6	0.003 **
ASM (kg/m^2^)	5.12 ± 0.84	4.73 ± 0.62	5.24 ± 0.86	0.005 **
Grip Strength (kg)	17.3 ± 5.1	14.7 ± 4.2	18.2 ± 5.2	0.003 **
Femur Neck BMD (g/cm^2^)	0.69 ± 0.11	0.66 ± 0.10	0.70 ± 0.11	0.094
Leptin (ng/mL)	23.1 ± 17.2	18.1 ± 14.5	25.2 ± 17.8	0.120
Compression Fracture (N)	38 (48.1%)	24 (30.4%)	14 (17.7%)	

BMI, body mass index; ASM, appendicular skeletal muscle mass; BMD, bone mineral density; Leptin, serum leptin concentration; Grip strength, handgrip strength (* *p* < 0.05, ** *p* < 0.01).

**Table 2 diagnostics-15-01620-t002:** Comparison between obese and non-obese groups.

Variable	Obese (Mean ± SD)	Non-Obese (Mean ± SD)	*p*-Value
Fat Index	11.6 ± 1.99	8.24 ± 1.76	<0.001 ***
Leptin (ng/mL)	31.8 ± 29.6	14.8 ± 10.5	<0.001 ***
ASM (kg)	5.37 ± 0.59	4.91 ± 0.46	<0.001 ***
Grip Strength (kg)	17.8 ± 4.63	16.8 ± 5.04	0.383
Femur Neck BMD (g/cm^2^)	0.599 ± 0.09	0.577 ± 0.06	0.259

Obesity was defined as BMI ≥ 25 kg/m^2^. BMI, body mass index; ASM, appendicular skeletal muscle mass; BMD, bone mineral density; Fat %, total body fat percentage; Fat Index, fat mass/height^2^; Leptin, serum leptin concentration; (* *p* < 0.05, *** *p* < 0.001).

**Table 3 diagnostics-15-01620-t003:** Correlation matrix between leptin, BMI, fat percentage, ASM, grip strength, and femur neck BMD.

Variable	BMI (r, 95% CI)	Fat Index (r, 95% CI)	Total Fat % (r, 95% CI)	ASM (r, 95% CI)	Grip Strength (r, 95% CI)	Femur Neck BMD (r, 95% CI)
Leptin	0.486 *** (0.297–0.638)	0.518 *** (0.335–0.663)	0.526 *** (0.345–0.669)	0.206 (−0.016–0.409)	−0.041 (−0.260–0.182)	−0.195 (−0.027–0.399)
BMI	—	0.892 *** (0.836–0.930)	0.755 *** (0.641–0.837)	0.529 *** (0.348–0.671)	0.222 * (0.001–0.422)	0.339 * (0.128–0.521)
Fat Index	0.892 *** (03.836–0.930)	—	0.902 *** (0.851–0.937)	0.468 *** (0.276–0.625)	0.247 * (0.027–0.444)	0.365 *** (0.157–0.543)
Total Fat %	0.755 *** (0.641–0.837)	0.902 *** (0.851–0.937)	—	0.144 (−0.079–0.354)	0.211 (−0.010–0.413)	0.293 *** (0.077–0.483)
ASM	0.529 *** (0.348–0.671)	0.468 *** (0.276–0.625)	0.144 (−0.079–0.354)	—	0.275 * (0.066–0.474)	0.283 * (0.066–0.474)
Grip Strength	0.222 * (0.001–0.422)	0.247 * (0.027–0.444)	0.211 (−0.010–0.413)	0.275 * (0.066–0.474)	—	0.185 (−0.037–0.390)

BMI, body mass index; ASM, appendicular skeletal muscle mass; BMD, bone mineral density; Fat %, total body fat percentage; Fat Index, fat mass/height^2^; Leptin, serum leptin concentration; CI, confidence intervals; (* *p* < 0.05, *** *p* < 0.001).

**Table 4 diagnostics-15-01620-t004:** Binomial logistic regression analysis for predicting fracture risk.

Predictor	Estimate	SE	Z	*p*-Value
Leptin	−0.0203	0.0132	−1.537	0.124
ASM	−0.8340	0.4360	−1.920	0.055
Grip Strength	−0.1010	0.0503	−2.000	0.046 *
Femur Neck BMD	−10.4000	3.6000	−2.890	0.004 **

ASM, appendicular skeletal muscle mass; BMD, bone mineral density; SE, standard error. (* *p* < 0.05, ** *p* < 0.01).

## Data Availability

The dataset used and/or analyzed during the current study is available from the corresponding author upon reasonable request.

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
