# Peer review of "Association of Leptin in Sarcopenia and Bone Density in Elderly Women: An Observational Analysis"

_diagnostics, 2025, doi:10.3390/diagnostics15131620_

Round 1
Reviewer 1 Report
Comments and Suggestions for Authors
The manuscript titled "Association of Leptin in Sarcopenia and Osteoporosis among Older Women: An Observational Analysis" by Dong Gyu Lee and Jong Ho Lee addresses a highly relevant yet relatively unexplored area in geriatric health research, focusing on the potential role of leptin in musculoskeletal health among older women. However, I have a number of concerns regarding the study’s design, methodological clarity, data presentation, and the interpretation of results. These issues should be addressed to improve the scientific rigor and overall impact of the manuscript.
Major Concerns
- Study Design: The authors employ a cross-sectional, observational analysis to explore associations between serum leptin, muscle mass/strength, and bone mineral density (BMD) in community-dwelling older women (n = 79). While appropriate for hypothesis generation, the cross-sectional design inherently limits causal inference. A longitudinal or case–control design would be better suited to clarify the temporal relationships between leptin dynamics and musculoskeletal outcomes.
- The data are not adjusted for important potential confounders, including comorbidities. This omission limits the interpretability and generalizability of the findings.
- The manuscript does not clearly define osteoporosis using standard diagnostic thresholds (e.g., T-score ≤ −2.5 at the hip or spine). This lack of diagnostic clarity is a significant limitation.
- Title: The inclusion of the term “osteoporosis” in the title is not justified. The study does not include a stratified analysis based on a clinical diagnosis of osteoporosis and instead focuses on continuous measures such as BMD and fracture risk. A more accurate and specific title—for example, emphasizing bone mineral density or vertebral fracture risk—would better reflect the scope and design of the study.
- Ethical Compliance: Although the manuscript states that IRB approval was obtained and informed consent collected, there is no reference to the Declaration of Helsinki or to the specific version of the declaration under which ethical review was conducted. Including this information is important to demonstrate compliance with internationally accepted standards for human research.
Methods Section:
- Serum leptin was measured by ELISA, but key details are missing. The manufacturer’s name, catalog and lot number, assay sensitivity, and intra-/inter-assay coefficients of variation should be provided. These details are necessary to ensure reproducibility and to assess assay reliability.
- The manuscript does not specify the statistical software or version used for the analyses. Additionally, justification for the covariates included in the logistic regression model is not provided. These omissions should be addressed.
- Effect sizes are presented without 95% confidence intervals, and key correlation results also lack confidence intervals. Providing this information would strengthen the statistical interpretation and credibility of the findings.
- Table 2: Several variables, including leptin and ASM, are labeled as “Higher / Lower” without numeric mean ± SD values. This format is unclear and should be corrected.
- Table 1: Frequencies of fractures in the comparison groups used to assess the association between leptin and fracture risk are not reported. This information is essential to contextualize the logistic regression results.
- The discussion would benefit from a more in-depth consideration of the potential mechanisms underlying age-related leptin resistance and its implications for sarcopenia and bone health. The role of unmeasured inflammatory markers (e.g., IL-6, TNF-α) should be acknowledged as a potential mediator and confounder.
- While the authors appropriately acknowledge the limitations of the cross-sectional design, some statements in the discussion suggest more robust associations than the data support. These should be tempered to reflect the observational nature and statistical limitations of the study.
Author Response
The manuscript titled "Association of Leptin in Sarcopenia and Osteoporosis among Older Women: An Observational Analysis" by Dong Gyu Lee and Jong Ho Lee addresses a highly relevant yet relatively unexplored area in geriatric health research, focusing on the potential role of leptin in musculoskeletal health among older women. However, I have a number of concerns regarding the study’s design, methodological clarity, data presentation, and the interpretation of results. These issues should be addressed to improve the scientific rigor and overall impact of the manuscript.
Response: We sincerely thank the Reviewer for the thoughtful and constructive feedback on our manuscript. We appreciate your detailed critique, which has helped us substantially improve the clarity, rigor, and quality of the manuscript. Below, we address each of your comments point by point. Changes made to the revised manuscript have been highlighted in red text.
Major Concerns
- Study Design: The authors employ a cross-sectional, observational analysis to explore associations between serum leptin, muscle mass/strength, and bone mineral density (BMD) in community-dwelling older women (n = 79). While appropriate for hypothesis generation, the cross-sectional design inherently limits causal inference. A longitudinal or case–control design would be better suited to clarify the temporal relationships between leptin dynamics and musculoskeletal outcomes.
Response: As the Reviewer rightly pointed out, cross-sectional analysis has inherent limitations in assessing interval changes within individuals and therefore may not fully capture the relationship between leptin and musculoskeletal parameters such as bone and muscle. Nevertheless, we attempted to minimize potential bias by restricting our analysis to women aged 65 years and older, thereby controlling for sex and age-related variation. Despite these efforts, we acknowledge that the limitation remains, and we have addressed this explicitly in the revised “Limitations” section of the manuscript.
2. The data are not adjusted for important potential confounders, including comorbidities. This omission limits the interpretability and generalizability of the findings.
Response: We appreciate the Reviewer’s valuable observation regarding the omission of comorbidity data as potential confounders. We agree that the presence of comorbid conditions (e.g., diabetes, cardiovascular disease, or chronic inflammatory disorders) may influence both leptin levels and musculoskeletal health. However, each comorbidity can exert independent effects on leptin concentrations, and accounting for all such variables typically requires large-scale population-based analyses. In our study, participants with severe or uncontrolled systemic illnesses were not included, minimizing the influence of major confounding conditions. Moreover, given that chronic conditions such as diabetes or persistent inflammation are known contributors to sarcopenia [1,2], their presence could potentially strengthen—rather than obscure—the associations between leptin, muscle mass, and bone density, which are the primary focus of this study.
[1]. Chhetri, Jagadish K., et al. "Chronic inflammation and sarcopenia: A regenerative cell therapy perspective." Experimental gerontology103 (2018): 115-123.
[2].Mesinovic, Jakub, et al. "Sarcopenia and type 2 diabetes mellitus: a bidirectional relationship." Diabetes, metabolic syndrome and obesity: targets and therapy(2019): 1057-1072.
3. The manuscript does not clearly define osteoporosis using standard diagnostic thresholds (e.g., T-score ≤ −2.5 at the hip or spine). This lack of diagnostic clarity is a significant limitation.
Response: We analyzed the presence or absence of osteoporosis in our dataset. However, when using the standard diagnostic threshold of T-score ≤ −2.5 at the hip to define osteoporosis, we found no statistically significant differences in factors such as leptin, ASM, or grip strength between osteoporotic and non-osteoporotic groups. While these factors tended to be lower in participants with osteoporosis, the differences did not reach statistical significance, which may be due to the age of our study population (≥65 years). Therefore, instead of using a binary osteoporosis diagnosis, we employed continuous bone mineral density (BMD) values as analytical variables. Among elderly individuals, even in the absence of overt osteoporosis, reduced muscle mass has been associated with increased fracture risk [1]. Thus, our study utilized BMD as a continuous variable, analyzed alongside ASM, to better capture the nuanced relationships with musculoskeletal health.
[1] Lee, Dong Gyu, and Jae Hwa Bae. "Fatty infiltration of the multifidus muscle independently increases osteoporotic vertebral compression fracture risk." BMC Musculoskeletal Disorders1 (2023): 508.
4. Title: The inclusion of the term “osteoporosis” in the title is not justified. The study does not include a stratified analysis based on a clinical diagnosis of osteoporosis and instead focuses on continuous measures such as BMD and fracture risk. A more accurate and specific title—for example, emphasizing bone mineral density or vertebral fracture risk—would better reflect the scope and design of the study.
Response: We appreciate the Reviewer’s insightful suggestion regarding the title of our manuscript. As noted, our analysis did not involve stratification based on a formal diagnosis of osteoporosis, but instead evaluated continuous bone mineral density (BMD) values and fracture risk. In response to this comment, we have revised the manuscript title to more accurately reflect the scope and design of the study. The new title is: “Association of Leptin with Sarcopenia and Bone Density in Elderly Women: An Observational Analysis”This revised title better represents the variables assessed and removes any potential confusion related to the diagnostic classification of osteoporosis.
5. Ethical Compliance: Although the manuscript states that IRB approval was obtained and informed consent collected, there is no reference to the Declaration of Helsinki or to the specific version of the declaration under which ethical review was conducted. Including this information is important to demonstrate compliance with internationally accepted standards for human research.
Response: The Declaration of Helsinki guideline is described in the Institutional Review Board Statement.
Methods Section:
1. Serum leptin was measured by ELISA, but key details are missing. The manufacturer’s name, catalog and lot number, assay sensitivity, and intra-/inter-assay coefficients of variation should be provided. These details are necessary to ensure reproducibility and to assess assay reliability.
Response: We appreciate the Reviewer’s comment regarding the need for detailed assay information. Serum leptin concentrations were measured using a commercially available ELISA kit: Human Leptin ELISA Kit (Millipore, Catalog No. EZHL-80SK). According to the manufacturer’s specifications, the intra-assay coefficient of variation ranged from 1.4% to 4.9%, and the inter-assay coefficient of variation ranged from 1.3% to 8.6%, depending on leptin concentration. These values have been added to the revised Methods section to ensure transparency and reproducibility.
2. The manuscript does not specify the statistical software or version used for the analyses. Additionally, justification for the covariates included in the logistic regression model is not provided. These omissions should be addressed.
Response: We thank the Reviewer for this helpful comment. The statistical analyses were conducted using IBM SPSS Statistics for Windows, Version 25.0 (IBM Corp., Armonk, NY, USA). We have now included this information in the Methods section of the revised manuscript. Regarding the logistic regression model, we selected covariates based on both clinical relevance and prior literature suggesting their association with fracture risk in elderly populations. Specifically, serum leptin, appendicular skeletal muscle mass (ASM), grip strength, and right hip BMD [1-3] were included as independent variables because they represent the key musculoskeletal and metabolic parameters investigated in this study. These covariates were chosen to evaluate their independent contributions to the likelihood of fracture occurrence. We have now clearly stated and justified these selections in the Methods section.
[1] Hong, Changbin, et al. "Body composition and osteoporotic fracture using anthropometric prediction equations to assess muscle and fat masses." Journal of cachexia, sarcopenia and muscle6 (2021): 2247-2258.
[2] Kamiya, Kuniyasu, et al. "Association between hand-grip strength and site-specific risks of major osteoporotic fracture: results from the Japanese Population-based Osteoporosis Cohort Study." Maturitas130 (2019): 13-20.
[3] Stone, Katie L., et al. "BMD at multiple sites and risk of fracture of multiple types: long‐term results from the Study of Osteoporotic Fractures." Journal of bone and mineral research11 (2003): 1947-1954.
3. Effect sizes are presented without 95% confidence intervals, and key correlation results also lack confidence intervals. Providing this information would strengthen the statistical interpretation and credibility of the findings.
Response: We sincerely appreciate the Reviewer’s insightful comment regarding the inclusion of 95% confidence intervals (CIs) for both effect sizes and correlation coefficients. In response, we have revised the relevant tables to incorporate 95% CIs for all correlation analyses. We believe these additions enhance the statistical rigor and transparency of the manuscript, thereby improving the clarity and interpretability of our findings.
Specifically:
Table 3 (correlation matrix) has been updated to present Pearson’s r values accompanied by their respective 95% confidence intervals.
4. Table 2: Several variables, including leptin and ASM, are labeled as “Higher / Lower” without numeric mean ± SD values. This format is unclear and should be corrected.
Response: We have revised Table 2 to present quantitative values in the format of mean ± standard deviation (SD) for all listed variables, including leptin and ASM.
5. Table 1: Frequencies of fractures in the comparison groups used to assess the association between leptin and fracture risk are not reported. This information is essential to contextualize the logistic regression results.
Response: We appreciate the Reviewer’s helpful suggestion regarding the inclusion of fracture frequencies in the comparison groups. We fully agree that this information is essential for contextualizing the logistic regression analysis and enhancing the interpretability of our findings. Accordingly, we have revised Table 1 to report the number and percentage of participants with compression fractures in both the sarcopenia and non-sarcopenia groups.
6. The discussion would benefit from a more in-depth consideration of the potential mechanisms underlying age-related leptin resistance and its implications for sarcopenia and bone health. The role of unmeasured inflammatory markers (e.g., IL-6, TNF-α) should be acknowledged as a potential mediator and confounder.
Response: We appreciate the Reviewer’s insightful suggestion to expand the discussion on the underlying mechanisms of age-related leptin resistance and its implications for sarcopenia and bone health. In the revised manuscript, we have added a more detailed discussion of leptin resistance in older adults, including how selective leptin resistance may contribute to impaired muscle function despite preserved fat mass. We also addressed the conflicting evidence regarding leptin’s dual role in muscle mass preservation and functional decline, citing relevant literature.
We agree that the absence of inflammatory markers (e.g., IL-6, TNF-α) is a limitation of our study. These cytokines may act as mediators or confounders in the relationship between leptin and musculoskeletal health. Furthermore, as part of our ongoing research, we are planning a follow-up study to explore the interplay between sarcopenia and immune-inflammatory responses in elderly individuals, which we hope will address these mechanistic gaps in more depth.
7. While the authors appropriately acknowledge the limitations of the cross-sectional design, some statements in the discussion suggest more robust associations than the data support. These should be tempered to reflect the observational nature and statistical limitations of the study.
Response: We appreciate the Reviewer’s insightful observation. As suggested, we have thoroughly reviewed the discussion section and carefully revised any overstatements that may have implied causal relationships beyond the scope of our cross-sectional design. We have explicitly clarified the observational nature of our study and tempered our interpretations to align with the limitations of the design and data.
Reviewer 2 Report
Comments and Suggestions for Authors
Firstly, I would like to thank the editor for giving me the opportunity to revise this paper. The manuscript examines the relationship between serum leptin levels and musculoskeletal health indicators in an older female cohort. The study addresses a relevant clinical topic and provides useful descriptive and analytical data. While the paper is clearly written and presents interpretable results, there are several structural, methodological and interpretative limitations that must be addressed to enhance its scientific value.
Firstly, despite using logistic regression, the analysis does not appear to control for important confounding variables, such as: physical activity; nutritional status; inflammatory markers; and comorbidities (e.g. diabetes, cardiovascular disease). This omission weakens causal inference and may explain some of the null associations reported.
Furthermore, the sample size is small and the study is cross-sectional. The study population is restricted to older Korean women attending a spine clinic, limiting applicability to broader, more diverse populations. Including a graphical abstract or a diagram illustrating the main findings and hypothesis (e.g. the relationship between leptin and fracture risk) would greatly improve clarity.
In conclusion, I think the paper can only be accepted after major revisions aimed at strengthening the analysis and interpretation of these data (through other parameters and confounders, if available) , and at better highlighting the identified limitations.
Author Response
Firstly, I would like to thank the editor for giving me the opportunity to revise this paper. The manuscript examines the relationship between serum leptin levels and musculoskeletal health indicators in an older female cohort. The study addresses a relevant clinical topic and provides useful descriptive and analytical data. While the paper is clearly written and presents interpretable results, there are several structural, methodological and interpretative limitations that must be addressed to enhance its scientific value.
Response: We sincerely thank the Reviewer for the thorough and constructive evaluation of our manuscript. We appreciate the acknowledgment of the study’s relevance and clarity. In response to your comments, we have carefully revised the manuscript to address the methodological and interpretative limitations identified. We have made clarifications in the discussion regarding the limitations of our cross-sectional design and small sample size, and provided further explanation of our analytical approach to enhance the scientific rigor and transparency of our findings.
1. Firstly, despite using logistic regression, the analysis does not appear to control for important confounding variables, such as: physical activity; nutritional status; inflammatory markers; and comorbidities (e.g. diabetes, cardiovascular disease). This omission weakens causal inference and may explain some of the null associations reported.
Response: We sincerely appreciate the Reviewer’s thoughtful critique regarding potential confounding variables such as physical activity, nutritional status, and comorbid conditions. We fully agree that these factors may influence both leptin levels and musculoskeletal health outcomes. Although we did not directly measure physical activity or nutrition, we used grip strength as a surrogate marker of muscle function and frailty, which has been widely validated in geriatric sarcopenia assessments.
Furthermore, as the Reviewer rightly pointed out, cross-sectional analysis has inherent limitations in assessing interval changes within individuals and therefore may not fully capture the relationship between leptin and musculoskeletal parameters such as bone and muscle. Nevertheless, we attempted to minimize potential bias by restricting our analysis to postmenopausal women aged 65 years and older, thereby controlling for sex and age-related variation. This limitation is explicitly acknowledged in the revised “Limitations” section.
Regarding comorbidities, we excluded participants with active malignancy, end-stage renal disease, or other severe systemic illnesses to minimize their confounding influence. While we recognize that conditions such as diabetes, cardiovascular disease, and chronic inflammation can independently affect leptin levels, comprehensive adjustment for all such variables would require a larger, population-based dataset. Notably, prior studies indicate that some of these conditions—such as diabetes and inflammation—may strengthen rather than obscure the associations under investigation [1,2].
These considerations have been incorporated into the revised Discussion to ensure that interpretations remain cautious and scientifically balanced.
References:
- Chhetri JK, et al. Chronic inflammation and sarcopenia: A regenerative cell therapy perspective. Exp Gerontol. 2018;103:115–123.
- Mesinovic J, et al. Sarcopenia and type 2 diabetes mellitus: a bidirectional relationship. Diabetes Metab Syndr Obes. 2019;12:1057–1072.
2. Furthermore, the sample size is small and the study is cross-sectional. The study population is restricted to older Korean women attending a spine clinic, limiting applicability to broader, more diverse populations. Including a graphical abstract or a diagram illustrating the main findings and hypothesis (e.g. the relationship between leptin and fracture risk) would greatly improve clarity.
Response: We agree with the Reviewer that a larger and more diverse sample would enhance the generalizability and strength of the study. However, in clinical settings, it is often difficult to obtain grip strength measurements in patients with fractures of the femur or upper limbs, which limits inclusion of broader patient populations. Additionally, our institution is a tertiary referral hospital, where patients commonly present with severe comorbidities. Including such individuals may amplify confounding effects and introduce significant bias. Therefore, we deliberately limited our study population to community-dwelling older women without major comorbidities, to better isolate the relationship between leptin, bone, and muscle health.
Furthermore, we believe that the relationship between leptin and specific comorbidities is best explored in disease-targeted populations. The strength of the present study lies in its focus on relatively healthy older adults, providing insights into leptin’s association with musculoskeletal health in the absence of overt disease-related confounders.
3. In conclusion, I think the paper can only be accepted after major revisions aimed at strengthening the analysis and interpretation of these data (through other parameters and confounders, if available) , and at better highlighting the identified limitations.
Response: We sincerely thank the Reviewer for their thoughtful and constructive feedback. In response to the concern that the manuscript required major revisions, we undertook a careful and comprehensive revision process to address the limitations previously identified. Furthermore, we revised the statistical reporting to include 95% confidence intervals and rephrased several sections of the Discussion and Conclusions to avoid overstating our findings, in accordance with the observational and cross-sectional nature of the study. We believe that these revisions have significantly strengthened the overall rigor and transparency of the manuscript, and we hope that the revised version now meets the expectations for publication.

Round 2
Reviewer 1 Report
Comments and Suggestions for Authors
The authors have, for the most part, addressed the reviewer’s comments in a thorough and constructive manner. The revised manuscript shows clear improvements in methodological transparency, data presentation, and interpretative caution. Most of the major concerns have been adequately resolved.
However, while the authors state that the study was conducted in accordance with the Declaration of Helsinki, the specific version or year of the revision (e.g., the 2013 Fortaleza revision) is not indicated, please add.
Author Response
We sincerely appreciate the Reviewer’s recognition of the improvements made in our revised manuscript and thank you for your thoughtful and constructive feedback throughout the review process.
In response to your final comment regarding the Declaration of Helsinki, we have updated the ethical compliance statement in the manuscript to specify the version and year of revision.
This revision has been reflected in the manuscript's Institutional Review Board Statement. We hope this adequately addresses your comment, and we thank you again for your valuable guidance.
The sentence now reads:
“The study was conducted in accordance with the ethical principles outlined in the Declaration of Helsinki (7th revision, Fortaleza, 2013) and was approved by the Institutional Review Board of Yeungnam University Hospital.”
Reviewer 2 Report
Comments and Suggestions for Authors
Dear Editor, Thank you for the opportunity to review the revised manuscript.
I would also like to thank the authors for the thoughtful and accurate revision of their work in response to the comments provided. The manuscript has been significantly improved and, in its current form, is suitable for publication.
Best regards,
PG
Author Response
We sincerely thank the Reviewer for the kind and encouraging comments. We are grateful for your constructive feedback and thoughtful suggestions, which greatly contributed to improving the clarity, rigor, and overall quality of our manuscript.
We are pleased that the revised version meets your expectations and appreciate your support in recommending the manuscript for publication.